# A 2-Year Retrospective Case Series on Isolates of the Emerging Pathogen *Actinotignum schaalii* from a Canadian Tertiary Care Hospital

**DOI:** 10.3390/microorganisms10081608

**Published:** 2022-08-09

**Authors:** Pramath Kakodkar, Camille Hamula

**Affiliations:** Department of Pathology and Laboratory Medicine, University of Saskatchewan, Saskatoon, SK S7N 0W8, Canada

**Keywords:** *Actinotignum schaalii*, *Actinobaculum schaalii*, urogenic infections, non-urogenic infections, ulcers

## Abstract

(1) Background: *Actinotignum schaalii* is an emerging, opportunistic pathogen often linked with UTIs but can extend beyond the urogenital system. Data on the clinical significance of *A. schaalii* are still emerging. (2) Methods: A retrospective review of *A. schaalii* isolates in a Canadian tertiary care hospital was conducted. The clinical data of patients that grew *A. schaalii* from January 2020 to 2022 were documented. Demographics, site, management, and microbiological parameters surrounding culture and sensitivities were recorded. (3) Results: A total of 43 cases of *A. schaalii* were identified. *Actinotignum schaalii* was primarily involved in UTIs (*n* = 17), abscesses (*n* = 9), bacteremia (*n* = 6), septic arthritis (*n* = 5), and ulcers (*n* = 5). *A. schaalii* had a slight predilection for polymicrobial infections (51.1%, *n* = 22 out of 43), with *Aerococcus urinae* (*n* = 5) being the most common coisolate. Susceptibility testing was only performed in two cases that showed sensitivity to beta-lactam antibiotics and resistance to metronidazole and ciprofloxacin. Amoxicillin–clavulanate (*n* = 5) is the most frequently prescribed antibiotic. (4) Conclusions: The non-urogenic clinical significance of *A. schaalii* remains undervalued. The management of *A. schaalii* infection is multimodal, consisting predominantly of antimicrobials and surgical procedures specific to the etiology. Clinicians should request sensitivities for *A. schaalii* so that appropriate antimicrobial coverage can be provided.

## 1. Introduction

*Actinotignum schaalii* has been recognized in recent years as an emerging uropathogen, but its nomenclature history remains checkered with taxonomic reclassification since its inception in 1997 as *Actinobaculum schaalii* [1,2]. Since 2015, the *Actinobaculum* genus diversified into two distinct genera: *Actinobaculum* and *Actinotignum*. The independent genus Actinobaculum is now composed of *Actinobaculum suis*, *Eubacterium suis,* and *Actinomyces suis* [3]. Contrastingly, the Actinotignum genus comprises *Actinotignum urinale*, *Actinotignum sanguinis*, and *Actinotignum schaalii* [4].

*A. schaalii* is a Gram-positive coccobacillus and a facultative anaerobe that is native to the urinary microbiota [5,6,7,8]. The current literature reveals that *A. schaalii* has been implicated in over 172 infections with a higher predilection for urinary tract infections (UTIs) (*n* = 121, 70%), bacteremia (*n* = 33, 19%), and abscesses (*n* = 12, 7%) [3]. Historically, the clinical underestimation of *A. schaalii* was linked to its lack of differentiation from *Actinomyces* and *Actinobaculum* spp., the absence of matrix-assisted laser desorption/ionization time-of-flight mass spectrometry (MALDI-TOF MS) databases, and suboptimal recovery and slow growth in routine cultures. Current clinical microbiology laboratories utilize MALDI-TOF MS and its accurate bacterial identification repository. Despite this, *A. schaalii* infections remain underdiagnosed, particularly when isolated in mixed culture scenarios. The role of *A. schaalii* in abscess formation is poorly understood. The absence of anaerobic culture media plates in routine urine workups also likely contributes to the decreased recognition of this pathogen as a cause of UTIs. A retrospective case series on patients infected with *Actinotignum schaalii* was conducted. The aim of this case series was to glean more insight into the sites of presentation, sources of infection, and other microbiological parameters influencing the spread of *Actinotignum schaalii* in our hospital. Additionally, the presence or absence of antimicrobial sensitivity requests by clinicians or reflex testing for *A. schaalii* sensitivities by the laboratory were evaluated to determine the perceived clinical significance of *A. schaalii*.

## 2. Materials and Methods

In this study, we reviewed the retrospective clinical data on patients at Royal University Hospital (RUH) in Saskatoon, Saskatchewan, from January 2020 to January 2022. This is a tertiary care hospital with 407 beds, which also supports microbiology testing for the surrounding communities. All hospital inpatients and outpatients diagnosed with infections and a microbiological report positive for *Actinotignum schaalii* were included in the analysis. *A. schaalii* recovered in this study grew on either routine aerobic urine culture plates (blood agar and chromogenic agar), from aerobic or anaerobic blood culture bottles, or on wound culture workups with brucella blood agar and blood agar incubated under both aerobic and anaerobic conditions. Urine cultures in our lab are not incubated anaerobically. *A. schaalii* are not strict anaerobes; they are facultative anaerobes that prefer anaerobic growth conditions but can grow slowly under aerobic conditions. The lack of anaerobic culture workup in urines likely contributes to the under-identification of this pathogen as a cause of UTIs. *A. schaalii* was also identified via a matrix-assisted laser desorption–ionization time-of-flight mass spectrometry (MALDI-TOF MS) (Biomerieux VITEK-MS, Biomerieux, France). The collected data included demographic variables, site of presentation, management strategies, microbiological parameters surrounding culture, antimicrobial sensitivities, and antimicrobials at the time of discharge. The results were analyzed on the SPSS 25 software. Patient progress notes were checked to record if clinicians requested antimicrobial sensitivities or guidance pertaining to the detection of *A. schaalii* in the microbiology reports.

We also performed a literature review on the MEDLINE database from January 1997 to November 2021, using the keywords “*Actinobaculum schaalii*”, and “*Actinotignum schaalii*”. This search yielded 55 articles on *A. schaalii* infections in the English language. In total, 30 publications were included in the literature review (166 patients). Articles with pooled data and no individual clinical data were excluded from our analysis. Ultimately, 12 articles were eliminated, as there was no usable primary clinical data for individual patients. Seven articles were eliminated, as they were review articles. The remaining six were not accessible.

## 3. Results

### 3.1. Patient Demographics

From January 2020 to January 2022, 43 patients with positive results for Actinotignum schaalii were identified. The inpatient (*n* = 32) and outpatient (*n* = 11) cohorts contributed to the overall population. The overall population distribution was right modal with a median age of 74 years (range: 12 to 94 years). A. schaalii had a 2.3-fold predilection for female patients (F:M = 30:13). Only one pediatric outpatient tested positive for A. schaalii. The length of stay duration for the inpatient population had a median of 5.5 days (range: 0 to 137 days).

### 3.2. Patient Comorbidity Profile

Table 1 summarizes the clinical diagnosis and the medical and surgical comorbidities in the overall population. The predominant type of infections from *A. schaalii* in this cohort were urinary tract infections (UTIs) (39.5%, *n* = 17 out of 43), and abscesses (20.9%, *n* = 9 out of 43). In patients with UTIs, the most common medical comorbidities included dementia or neurological conditions leading to urinary retention. The leading site for abscesses was the breast (44.4%, *n* = 4 out of 9), and the corresponding medical history consisted of recurrent breast infections managed with incision and drainage and breast cancer patients on chemotherapy.

Appendix A summarizes the medical and surgical comorbidities in the overall population. Most patients had cardiologic comorbidities (*n* = 29) and endocrinopathies (*n* = 25) with hypertension (*n* = 16), type II diabetes mellitus (*n* = 9), dementia (*n* = 9), and GERD (*n* = 9) leading within each of these cohorts, respectively.

### 3.3. Management Prior to Microbial Culture

Table 2 shows a summary of the clinical diagnoses, management strategy, and antimicrobial choice prior to cultures. The abscesses were found in a variety of locations. Antimicrobial management with ceftriaxone (2 g for 3 days) and surgical management with incision and drainage were the popular strategies for abscess treatment. Patients diagnosed with urosepsis and UTIs were managed with a variety of antibiotic agents among which nitrofurantoin (200 mg for 7 days), ciprofloxacin (1 g for 7 days), and ceftriaxone (2 g for 10 days) were popular choices. Patients with septic arthritis were predominantly managed with daptomycin (6 mg/Kg/day for 42 days). Patients with ulcers, pyelonephritis, and toxic megacolon were primarily managed with surgical interventions such as debridement, nephrostomy, and subtotal colectomy, respectively.

Table 2 shows a summary of the sources tested in the microbiological laboratory to assist with the clinical diagnosis. Abscess cases were often supplemented with swabs (*n* = 6 out of 9). The cases of UTIs, urosepsis, and pyelonephritis sent their specimen sources as sterile catheter urine (*n* = 9 out of 23). Ulcer and septic arthritis cases were dominated by swabs (*n* = 7 out of 10) as their specimen source.

The antibiotic choices prior to culture were appropriate in most of the cases. Our literature review showed that the *A. schaalii* is resistant to metronidazole and ciprofloxacin; therefore, a few patients (*n* = 5) in the urosepsis and UTI groups did not have the appropriate initial antimicrobial coverage. The appropriate antibiotic de-escalation after receiving the culture results is discussed in Section 3.5.

### 3.4. Microbiology Laboratory Investigations

*A. schaalii* are facultatively anaerobic Gram-positive bacilli. Figure 1 collates the findings of the Gram staining test for samples sent for abscesses (Figure 1A), ulcers (Figure 1C), septic arthritis (Figure 1E), and cases involving urosepsis (Figure 1G). The Gram staining test was not performed in cases involving UTIs and pyelonephritis. The single case involving toxic megacolon showed abundant Gram-negative bacilli and Gram-positive cocci and bacilli. All the infections shown in Figure 1 had abundant polymorphonuclear leukocytes, and most of the cases had polymicrobial organisms. The Gram stain profiles in abscesses showed that many of the cases had few (*n* = 3) Gram-negative bacilli and scant Gram-positive bacilli (*n* = 3). The predominant Gram stain profiles for the ulcers had abundant to moderate staining for Gram-positive cocci (*n* = 4). The Gram stain profiles in urosepsis were mostly scant (*n* = 2) Gram-negative bacilli. The Gram stain profiles in septic arthritis predominantly showed abundant Gram-positive cocci (*n* = 2) and Gram-positive bacilli (*n* = 2).

Figure 1 also shows the collated results for aerobic and anaerobic cultures in the wound specimens. Most infections had failed growth in aerobic and mixed anaerobic cultures. Blood cultures were performed only in cases of urosepsis and pyelonephritis. A urine culture was performed in cases of UTIs (*n* = 9), pyelonephritis (*n* = 1), urosepsis (2), and abscess (*n* = 1). Urine isolates in which *A. schaalii* were isolated had pure anaerobic growth (*n* = 6) and mixed growth (*n* = 3). The rest of the cases had predominantly mixed anaerobic growth.

### 3.5. Antimicrobial Tailoring after Microbial Culture

Table 3 shows a summary of the sources tested in the microbiological laboratory to assist with the clinical diagnosis. There was an equal distribution between polymicrobial (51.1%, *n* = 22 out of 43) and monomicrobial (48.8%) *A. schaalii* infections. The polymicrobial cohort consisted of 20 coisolated organisms, most commonly *Aerococcus urinae* (*n* = 5). UTI cohorts were predominantly monomicrobial.

The clinical microbiology laboratory at RUH did not perform routine susceptibility testing for *A. schaalii* isolates prior to this review. The only two documented cases for which a clinician requested sensitivity for *A. schaalii* were a case of pyelonephritis and a case of urosepsis secondary to traumatic catheterization. The susceptibility testing for *A. schaalii* in the pyelonephritis case showed sensitivity to ceftriaxone, penicillin G, trimethoprim–sulfamethoxazole, vancomycin, and resistance to ciprofloxacin. Similarly, in the urosepsis case, sensitivity to penicillin and amoxicillin–clavulanic acid and resistance to metronidazole were observed. Both patients were discharged on ampicillin. *Actinotignum schaalii* was predominantly involved in polymicrobial infections. In the UTI cohort, there were 12 patients with pure isolates of *A. schaalii,* and 4 patients had less than 2 coisolated organisms with over 100,000 CFU/mL of *A. schaalii*. Similarly, septic arthritis (*n* = 2) and abscess (*n* = 2) cohorts had less than three coisolated organisms with over 100,000 CFU/mL enrichment of *A. schaalii*.

The most popular antimicrobial choice at discharge in the monomicrobial and polymicrobial cohorts was metronidazole (*n* = 3) and clindamycin (*n* = 3), respectively. In the overall cohort, amoxicillin–clavulanate (*n* = 5) was the popular antimicrobial agent. Only septic arthritis and perforated viscus required a longer mean antibiotic coverage duration of 29 ± 10 days. All other infections of *A. schaalii* required a mean antibiotic coverage duration of 8 ± 3 days.

We conducted a detailed literature review of *Actinotignum schaalii* infections. A summary of the non-urological infections caused by *A. schaalii* is shown in Table 4.

Non-urological *A. schaalii* infections had a 1.2-fold predilection for polymicrobial growth in comparison to the monomicrobial occurrence. *A. schaalii* was isolated with *Enterococcus faecalis* (*n* = 7), *Peptoniphilus asaccharolyticus* (*n* = 7), and *Aerococcus urinae* (*n* = 5). Contrastingly, urological infections had a 1.5-fold predilection for monomicrobial occurrence compared with polymicrobial growth. These findings were consistent in our case series. The most common types of infections in the non-urological group were abscesses, with the groin and breast being the most common sites. Interestingly, the breast was a common site of infection in our case series. It is also important to note that the abscess cohort had the youngest population, with a mean age of 38 ± 2.9 years.

A summary of the urological infections caused by *A. schaalii* is shown in Appendix A. UTI (*n* = 65 out of 76) was the most common infection in this cohort with 60.5% (*n*= 46 out of 76) of cases being monomicrobial. In the polymicrobial cases, the most common coisolates were *E. coli* (*n* = 6) and *A. urinae* (*n* = 5). The most common antibiotic prescribed in this cohort was amoxicillin (*n* = 16).

## 4. Discussion

*A. schaalii* is an emerging human pathogen predominantly starting to be recognized as a possible uropathogen. Microbiology laboratories still struggle with how to report this organism, particularly in non-urine sources and mixed culture scenarios. Our case series (*n* = 43) is the largest case series of *A. schaalii* infection reported in North America to date. The data collection process during the COVID-19 pandemic could have affected the admission rates, leading to a possible decrease in the incidence rate of *A. schaalii* detection. Additionally, our MALDI-TOF database did not include *A. schaalii* prior to 2020; hence, we did not search and construct a database with *A. schaalii* infections in an earlier time period. In our literature review (*n* = 136), the median age was 70 years (range: 0.66 to 101 years). There was a bimodal distribution (Appendix A) with a smaller peak in the 20–30 years age group and a larger peak in the 80–90 years age group. The majority of the patients were in the 60–90 age group. The overall cohort showed a 2.1-fold predilection for males (M:F= 92:44). Contrastingly, in our case series, there was a higher predilection for female patients (F:M = 30:13).

In our case series, *A. schaalii* infections were primarily involved in UTIs (*n* = 17), abscesses (*n* = 9), bacteremia (*n* = 6), and septic joint (*n* = 5). Similarly, the literature review showed that UTIs (*n* = 65), abscesses (*n* = 42), and bacteremia (*n* = 38) were the leading infections caused by *A. schaalii*. Interestingly, other than our case series (*n* = 5), there were no documented cases of ulcers infected by *A. schaalii* in the literature. The majority of the cases from our literature review were from France (23.5%), Denmark (16.2%), Greece (14.7%), and Sweden (14.7%). Interestingly, 83.8% of reported cases were from European countries, followed by North America (11.8%) and South America (2.9%). Only 40% (*n* = 21 out of 55) of the cases utilized MALDI-TOF-MS after 2013 to identify *A. schaalii*. With the increasing implementation of MALDI-TOF-MS in clinical microbiology, it is expected that routine labs will continue to see an increase in the number of *A. schaalii* isolates, particularly in mixed culture scenarios.

It cannot be discerned whether *A. schaalii* activates an overwhelming localized immune response to form abscesses or *A. schaalii* acts as a synergistic copathogen. Understanding the precise role of *A. schaalii* in the pathogenesis of abscess formation warrants further exploration. Future studies could incorporate and correlate these microbiological findings with the antibody serology against *A. schaalii* to explore the pathogenic role of this organism in abscesses. The role of *A. schaalii* in mixed infections is not always clear. As seen in our study, it can be coisolated along with known pathogens in many cases.

*A. schaalii* is a commensal organism and is found along the urogenital mucosa and urethral passage [26]. Additionally, in one study of patients undergoing extracorporeal shock wave lithotripsy, their preprocedural testing revealed that *A. schaalii* was present in the samples of urine (*n* = 14), urine and groin swabs (*n* = 7), and vaginal swabs (*n* = 6). However, no *A. schaalii* was present in any fecal samples [26].

Interestingly, our case series revealed that in UTI (*n* = 3) and urosepsis (*n* = 3) patients, there was a recent prior history of cystoscopies. Additionally, the literature review revealed that there were 13 cases of benign prostatic hyperplasia, and 4 had a documented history of procedures such as transurethral resection of the prostate. It is a possibility that both these procedures could help *A. schaalii* to form cystitis and pyelonephritis at sites beyond its native residence or in severe cases disseminate via the bloodstream in cases of septic joint infections, urosepsis, and endocarditis. However, given the small sample size, this finding needs to be considered with extreme caution.

From our case series, we found that a patient with a clinical history of hypertension (*n* = 16), dementia (*n* = 9), T2DM (*n* = 9), GERD (*n* = 9), and obesity (*n* = 8) has a propensity to have *A. schaalii* infections. These findings can be explained by the fact that most of the patients were elderly, and the above comorbidities tend to have a higher prevalence in this cohort. Despite this, the virulence factors of *A. schaalii* as well as its ability to cause local and systemic infections in patients with certain underlying endocrinopathies, immunosuppressive states, and cancers have not been elucidated. In our case series, both patients with pyelonephritis had a history of prostate cancer. Moreover, there were five cases of prostate cancer with *A. schaalii* UTI (*n* = 1), Fournier gangrene (*n* = 1) and bacteremia (*n* = 3) [13,16,18,20]. The literature review also showed four cases of bladder cancers with *A. schaalii* infections, with the non-muscle-invasive bladder cancer being the commonest type [5,9,13,27].

The management of *A. schaalii* is multimodal, consisting predominantly of antimicrobials and various surgical procedures depending on the site and etiology. All the cases in the literature review had sensitivities requested and the antimicrobials were appropriately de-escalated. Conversely, in our case series, empiric treatment was often prescribed rather than tailored therapy. The most popular antimicrobial choice at discharge in the monomicrobial and polymicrobial cohorts was metronidazole (*n* = 3) and clindamycin (*n* = 3), respectively. Overall, metronidazole (*n* = 4) and amoxicillin–clavulanate (*n* = 4) were the popular antimicrobials after positive speciation. We only had two documented cases (pyelonephritis and urosepsis) wherein a clinician requested sensitivity for *A. schaalii*. In these cases, *A. schaalii* was sensitive to penicillin and amoxicillin–clavulanate and resistant to metronidazole and ciprofloxacin. This suggests that many of the patients (18.6%, *n* = 8 out 43) treated with metronidazole and ciprofloxacin for anaerobic coverage did not have appropriate coverage for Actinotignum. In the literature review, the initial broad-spectrum antibiotics utilized in the cases of Fournier gangrene, Perineal hidradenitis suppurativa, Cauda equina abscess, and bacteremia were resistant. The literature showed that *A. schaalii* was sensitive to amoxicillin (*n* = 13), penicillin G (*n* = 12), piperacillin–tazobactam (*n* = 11), vancomycin (*n* = 11), and amoxicillin–clavulanate (*n* = 10). Contrastingly, *A. schaalii* was resistant to ciprofloxacin (*n* = 13), trimethoprim–sulfamethoxazole (*n* = 12), and metronidazole (*n* = 11). Since there is a prevalence of *A. schaalii* in the Province of Saskatchewan, it might be useful for microbiology labs to perform reflex testing for sensitivities. It will help clinicians optimize appropriate antimicrobials sooner and mitigate further antibiotic-resistant conditions.

## 5. Conclusions

*A. schaalii* is not just an emerging uropathogen, and its clinical significance remains undervalued. Although the majority of *A. schaalii* cultures are polymicrobial, clinicians should request sensitivities for *A. schaalii* so that appropriate antimicrobial coverage, as well as antimicrobial stewardship with de-escalation from broad-spectrum coverage, can be provided to manage this pathogen effectively.

## Figures and Tables

**Figure 1 microorganisms-10-01608-f001:**
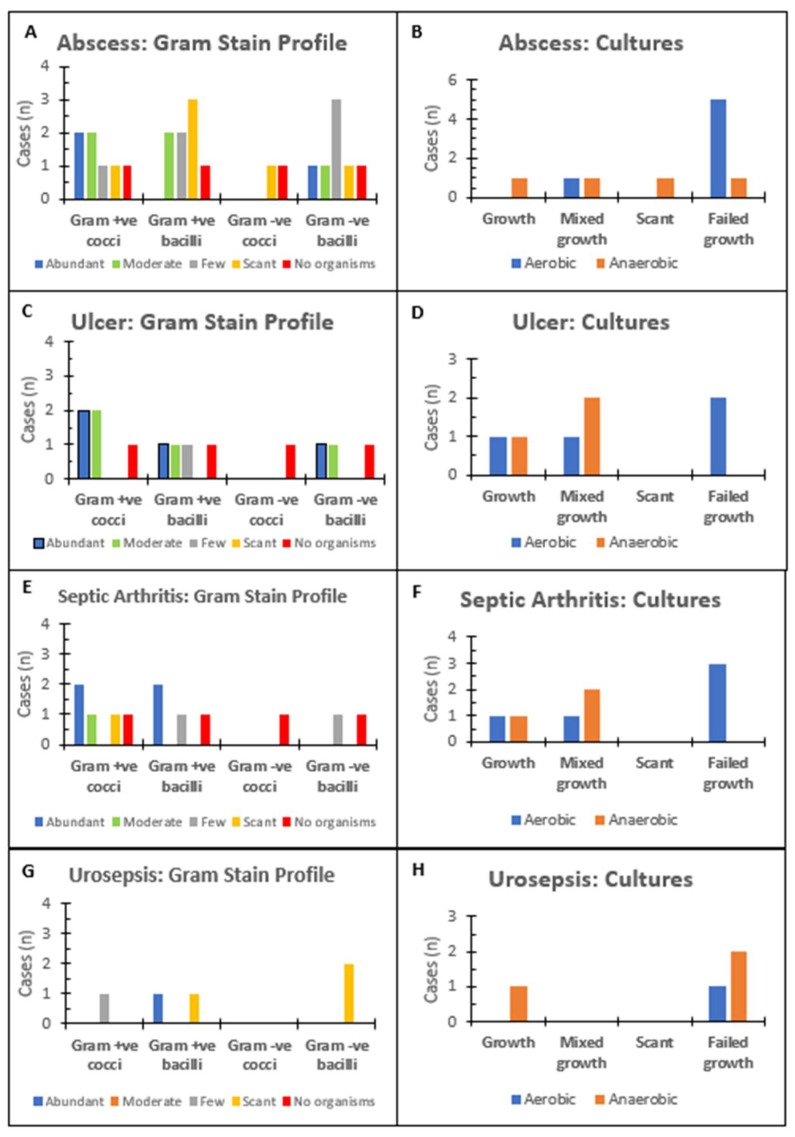
Summary of microbiology tests for samples sent for each of the infections: (**A**) Gram stain profile for abscess samples; (**B**) culture results for abscess samples; (**C**) Gram stain profile for ulcer samples; (**D**) culture results for ulcer samples; (**E**) Gram stain profile for septic arthritis samples; (**F**) culture results for septic arthritis samples; (**G**) Gram stain profile for urosepsis samples; and (**H**) culture results for urosepsis samples.

**Table 1 microorganisms-10-01608-t001:** Summary of clinical diagnosis, and the medical and surgical comorbidities in the inpatient population.

Clinical Diagnosis (*n*)	Site/Etiology (*n*)	Comorbidities (*n*)
Abscess (9)	Breast (4)	**Endocrine:** T2DM (2), obesity (2), SLE (1) **Gastrointestinal:** hemorrhoids (2), appendicitis (2), GERD (1) **Cardio:** HTN (3), HF (1) **Neuro/Psych:** anxiety (2), depression (2), spina bifida (1), idiopathic intracranial hypertension (1) **Breast:** Recurrent breast infection (2), breast cancer on chemotherapy (1) **Gynecology/GU:** Endometriosis (3) **Others:** CKD (1), OA/RA (1), COPD (1), DVT (1)
	Neck (1)	
	Pfannenstiel Line (1)	
	Periclitoral (1)	
	Umbilicus (1)	
	Thigh (1)	
Ulcer (5)	Toe base (post-amputation) (1)	**Endocrine:** T2DM (3), hypothyroidism (1) **Gastrointestinal:** GERD (2), IBD (1), hernia (1) **Cardio:** HTN (3), arrhythmia (2) **Neuro/Psych:** Dementia (2), TIA (1) **Others:** RA (1), peripheral vascular disease (1), recurrent foot infection (1)
	Diabetic foot ulcer (3)	
	Coccygeal pressure ulcer (1)	
Urosepsis (4)	Traumatic catheterization (1)	**Endocrine:** hypothyroidism (2), OSA (1) **Gastrointestinal:** GERD (2), hernia (1), cholecystitis (1), **Cardio:** HTN (2), arrhythmia (1), CAD (1) **Neuro/Psych:** Dementia (1), ET (1), CVA/TIA (1), CP (1) **Gynecology/GU:** Urolithiasis (2), BPH (2), UTIs (1), trabeculated hypotonic bladder (1) **Others:** COPD (2), CKD (1), OA/RA (1), DVT (1), anemia (1), knee replacement (1)
	Ureteric stone (2)	
	Catheterization of atonic bladder (1)	
UTIs (17)	Dementia (6)	**Endocrine:** Dyslipidemia (3), T2DM (1) **Cardio:** Arrhythmia (1), CAD (1) **Neuro/Psych:** Dementia (6), CVA/TIA (3), PD (1), NPH (1), vertigo (1), spina bifida (1), arachnoid cyst (1) **Gyne/GU:** UTIs (2), cystocele (2), BPH (1), endometriosis (1) **Others:** OA/RA (1), GERD (1)
	Acute urinary retention (4)	
	Cystocele (2)	
	CVA (3)	
Pyelonephritis (2)	Castration-resistant prostate cancer (1)	**Gynecology/GU:** Prostate cancer (2)
	Obstructive uropathy with ARF (1)	
Septic arthritis (5)	Hip (4)	**Endocrine:** T2DM (3), OSA (1), hyperthyroidism (1), hypothyroidism (1) **Cardio:** HTN (3), HF (3), aortic disease (2), CABG (1), arrhythmia (1) **Renal:** CKD (3), renal transplant (1) **Gastrointestinal:** GERD (2), diverticulitis (1), colitis (1), cirrhosis (1), chronic pancreatitis (1) **Orthopedic:** elective hip replacement (1), hardware-associated osteomyelitis (1), hip fracture (1) **Others:** Gout (2), COPD (1), Anemia (1), peripheral neuropathy (1), pulmonary embolism (1),
	Knee (1)	
Perforated Viscus (1)	Toxic Megacolon (1)	**Gastrointestinal:** Appendicitis (1), IBD (1), GERD (1) cholecystitis (1) **Hematology:** DVT (1), PE (1), anemia (1) Others: HTN (1)

**Abbreviations:** genitourinary (GU), type 2 diabetes mellitus (T2DM), systemic lupus erythematosus (SLE), hypertension (HTN), heart failure (HF), chronic kidney disease (CKD), osteoarthritis (OA), rheumatoid arthritis (RA), gastroesophageal reflux disease (GERD), chronic obstructive pulmonary disease (COPD), deep vein thrombosis (DVT), inflammatory bowel disease (IBD), transient ischemic attack (TIA), obstructive sleep apnea (OSA) coronary artery disease (CAD), cerebrovascular accident (CVA), cerebral palsy (CP), benign prostatic hyperplasia (BPH), urinary tract infection (UTI), Parkinson’s disease (PD), normal pressure hydrocephalus (NPH), acute renal failure (ARF), coronary artery bypass graft (CABG).

**Table 2 microorganisms-10-01608-t002:** Summary of the clinical diagnoses, lab specimens, management strategies, and antimicrobial choices prior to cultures.

Clinical Diagnosis (*n*) and Specimen	Site/Etiology (*n*)	Treatment Strategy (*n*)	Antimicrobials before Cultures (Dosage) (*n*)
**Abscess (9)** Specimen Swabs (6), Aspirate (3)	Breast (4)	Abx (3), I/D (2)	Daptomycin (IV 6 mg/kg OD for 2/52) (1), ceftriaxone (2 g for 3/7) (2)
	Neck (1)	Abx for HAP	Piperacillin–tazobactam (IV 2.25 mg q6h, 3/7) deescalated to azithromycin (PO 250 mg OD 3/7)
	Pfannenstiel Line (1)	I/D	NA
	Periclitoral (1)	I/D	NA
	Umbilicus (1)	I/D	NA
	Thigh (1)	Abx, I/D	Doxycycline (PO 100 mg BID 7/7)
**Ulcer (5)** Specimen Swabs (3), Tissue (2)	Toe base (post-amputation) (1)	Abx (1)	Piperacillin–tazobactam (IV 2.25 mg q6h)
	Diabetic foot ulcer (3)	Debridement (3)	NA
	Coccygeal pressure ulcer (1)	Repositioning (1)	NA
**Urosepsis (4)** Specimen Aspirate (3), Venipuncture (2)	Traumatic catheterization (1)	Abx (1)	Piperacillin–tazobactam (IV 4.5 g q6h)
	Ureteric stone (2)	Cystoscopy (2), stent (2), Abx (2)	Nitrofurantoin (100 mg BID 7/7) (1)
	Catheterization of atonic bladder (1)	Abx (1)	Ceftriaxone (IV 2 g for 4/7), ciprofloxacin (IV 500 mg bid 3/7)
**UTIs (17)** Specimen Sterile catheter (6), voided midstream (5), cystoscopy collected (3)	Dementia (6)	Abx (2), NA (4)	NA (2)
	Acute urinary retention (4)	Abx/self-catheterization (3), NA (1)	Ciprofloxacin (IV 500 mg bid 7/7), NA
	Cystocele (2)	Cystoscopy (2), Abx (1)	Nitrofurantoin (PO 400 mg bid 3/7)
	CVA (3)	Abx (1), NA (2)	Ceftriaxone (IV 2 g q12h) and metronidazole (IV 500 mg TID 2/52)
**Pyelonephritis (2)** Specimen Sterile catheter (1), Venipuncture (1)	Castration-resistant prostate cancer (1)	Nephrostomy (1)	NA
	Obstructive uropathy with ARF (1)	Bil. nephrostomy + ante stents (1)	NA
**Septic arthritis (5)** Specimen Aspirate (1), Swab (4)	Hip (4)	Abx (3), Debridement (1), THR (1), NA (1)	Cefazolin (1), daptomycin (IV 6 mg/kg OD for 6/52) (2), amoxiclav (PO 875–125 mg BID, 2/52) (1), TMP-SMX (IV 4 mg/kg OD for 8/52) (1)
	Knee (1)	Debridement (1)	NA (1)
**Perforated Viscus (1)** Specimen Aspirate (1),	Toxic Megacolon (1)	subtotal colectomy and end ileostomy	NA (1)

**Abbreviations:** antibiotics (Abx), incision and drainage (I/D), hospital-acquired pneumonia (HAP), cerebrovascular accident (CVA), bilateral (bil.), not applicable (NA), acute renal failure (ARF), total hip replacement (THR), trimethoprim–sulfamethoxazole (TMP-SMX), amoxicillin–clavulanate (amoxiclav).

**Table 3 microorganisms-10-01608-t003:** Summary of monomicrobial and polymicrobial culture results and subsequent antimicrobial choice.

Clinical Diagnosis (*n*)	Culture Type (*n*)	No. of Coisolates Mode [Range]	Organism Identity (*n*)	Antimicrobials After Cultures (Dosage) (*n*)
**Abscess (9)**	Mono (2)	NA	*Actinotignum schaalii* (2)	Metronidazole (250 mg, qid, 10/7) (1), cephalexin (500 mg, qid, 10/7) (1)
	Poly (7)	3 [2 to 4]	*A. schaalii* (7), *Propionibacterium avidum* (2), *Finegoldia magna* (1), *Escherichia coli* (1), *Peptoniphilus asaccharolyticus* (1), *Klebsiella pneumoniae* (1), *Morganella morganii* (1), *Staphylococcus hominis* (1), *Streptococcus intermedius* (1), *Actinomyces neuii* (1), *Bacillus circulans* (1), *Staphylococcus aureus* (1), *Staphylococcus epidermidis* (1), *Pseudomonas aeruginosa* (1)	Clindamycin (300 mg, qid, 7/7) (2), cefixime (400 mg, od, 14/7) (1), TMP (200 mg, bid, 7/7) (1), doxyclicine (100 mg, bid, 7/7) (1), amoxicillin–clavulanate (250 mg, tid, 10/7) (1)
**Ulcer (5)**	Mono (1)	NA	*A. schaalii* (1)	Cephalexin (500 mg, qid, 8/7) (1)
	Poly (4)	2 [2 to 5]	*A. schaalii* (4), *Enterococcus faecalis* (2), *Staphylococcus lugdunensis* (1), *S. aureus* (1), *P. aeruginosa* (1), *Bacteroides fragilis* (1), *Proteus mirabilis* (1), *S. epidermis* (1), *F. magna* (1), *Staphylococcus pettenkoferi* (1)	Clindamycin (300 mg, qid, 10/7) (1), amoxicillin–clavulanate (500–125 mg, bid, 14/7) (1)
**Pyelonephritis (2)**	Mono (1)	NA	*A. schaalii ** (1)	Ampicillin (2 g q6h, 10/7)
	Poly (1)	2	*A. schaalii ** (1), *Candida albicans* (1)	NA
**Septic Arthritis (5)**	Mono (3)	NA	*A. schaalii* (3)	TMP-SMX (800–160 mg, qid, 42/7) (1), cephalexin (500 mg, qid, 4/7) (2), NA (1)
	Poly (2)	2	*A. schaalii* (2), *F. magna* (1), *E. faecalis* (1)	Amoxicillin–clavulanate (875–125 mg, bid, 14/7) (1), NA (1)
**Perforated viscus (1)**	Poly (1)	2	*A. schaalii* (2), *B. fragilis* (1)	Ciprofloxacin (500 mg, bid, 30/7) (1), metronidazole (500 mg, bid, 30/7) (1)

* Antimicrobial sensitivities for *A. schaalii* were requested by the clinical team.

**Table 4 microorganisms-10-01608-t004:** Summary of literature review, monomicrobial and polymicrobial culture results, and subsequent management options in *A. schaalii*-related non-urological infections.

Clinical Diagnosis	Site/Etiology	Culture Results (*n*); Coisolates	Treatment Strategy (*n*)	Source
**Abscess (42)** Mean age = 38 ± 2.9, M:F (21:21) Aspirate (40), Blood (1), Tissue (1)	Abdominal (3)	Mono (2), Poly (1); CoNS	I/D (3), linezolid (2), pristinamycin (1)	[9,10]
	Breast (9)	*Mono (3), Poly (6); Streptococcus constellatus, Gemella haemolysans, CoNS, Escherichia coli, Klebsiella pneumoniae, Morganella morganii, Staphylococcus hominis, Propionibacterium avidum, Finegoldia magna, Peptoniphilus asaccharolyticus*	I/D (6), daptomycin (1), ceftriaxone (2), amoxiclav (2)	[10,11], *
	Fournier gangrene (3)	Mono (3)	I/D (1), vancomycin (1), ciprofloxacin (1), metronidazole (2), amoxiclav (2)	[11,12,13]
	Perineal hidradenitis suppurativa (3)	Poly (3); *Prevotella melaninogenica* (2), *Fusobacterium spp., Aerococcus spp.*	I/D (3), amoxiclav (1), clindamycin (1), minocycline (1)	[10]
	Pilonidal (3)	Mono (3)	I/D (1), cloxacillin (2), vancomycin (1)	[10,14]
	Vagina (3)	Mono (1), Poly (2); *Actinomyces turicensis, Bacteroides fragilis*	I/D (3), iodine (1)	[10], *
	Surgical site (2)	Poly (2); *Acinetobacter spp., Helcococcus spp., Anaerococcus spp., Actinomyces neuii*	I/D (1), amoxiclav (1)	[10], *
	Groin (12)	Mono (6), Poly (6); *Arcanobacterium pyogenes* (2), *Aerococcus sp., Enterococcus Faecalis* (2), *Propionibacterium avidum, Bacillus circulans*	I/D (10), amoxiclav (3), imipenem (2), pristinamycin (3), linezolid (2), doxycycline	[9,10], *
	Inguinal (1)	Mono (1)	I/D, cloxacillin	[14]
	Malleolus (1)	Poly (1); *Peptostreptococcus spp.*	I/D	[10]
	Neck (1)	Poly (1); *Streptococcus intermedius*	I/D, amoxiclav	*
	Cauda equina, Intradural (1)	Poly (1); Non-hemolytic streptococci	I/D, PCN, metronidazole	[5]
**Bacteremia (38)** Mean age = 71.7 ± 2.1, M:F (27:11), Aspirate (6), Blood (16), Urine (10), Urine and Blood (10)	Unknown source (9)	Mono (5), Poly (4); *Peptostreptococcus asaccharolyticus*, Mixed Anaerobe, Urethral flora, *streptococcus, Bacteroides fragilis*	Surgical management (3), CTX (3), cefixime, PIP-TZN (2), amoxiclav, ofloxacin, ciprofloxacin, metronidazole, daptomycin, cefepime, amoxicillin	[9,13,15,16,17], *
	Urosepsis (29)	Mono (10), Poly (19); *Enterococcus faecalis* (4), *Actinotignum urinale* (2), CoNS, *Aerococcus urinae* (3), Urethral flora (2), *Staphylococcus epidermidis, Prevotella, alpha-hemolytic streptococci, Actinomyces sp. and Peptostreptococcus anaerobius* (2), *Citrobacter braakii, Pseudomonas aeruginosa, Escherichia coli* (2), Non-hemolytic *Streptococcus, Aeromicrobium massiliense, peptostreptococcus asaccharolyticus* (2), *Proteus mirabilis*	PIP-TZM (6), metronidazole (3), cefuroxime (3), gentamicin (3), meropenem, ciprofloxacin (9), metronidazole, CTX (3), amoxicillin, amoxiclav (3), ampicillin, nitrofurantoin, TMP-SMX (2), cefotaxime (5), vancomycin, cefepime, levofloxacin, nephrectomy, nephrostomy catheter (2)	[5,9,13,15,16,18,19,20], *
**Septic Joint (6)** Mean age = 68.3 ± 4.5, M:F (3:3) Tissue (3), Aspirate (2), Disc Biopsy (1)	Incidental findings during elective hip replacement surgery (2)	Mono (1), Poly (1); Finegoldia magna	Debridement, cefazolin	*
	Hip hardware-associated osteomyelitis (3)	Mono (2), Poly (1); Enterococcus faecalis	Daptomycin (2), amoxiclav, TMP-SMX, amoxicillin, gentamicin, rifampin	[21], *
	Vertebral Osteomyelitis (1)	Poly (1); Corynebacterium striatum	Amoxiclav	[22]
**Infective Endocarditis (3)** Mean age = 68 ± 10.9, M:F (3:0) Blood (2), Urine and blood (1)	Native valve (2)	Mono (1), Poly (1); Escherichia coli	Amoxicillin (2), amoxiclav, gentamicin (2), aortic valve replacement surgery (2), tricuspid valve replacement	[23,24]
	Prosthetic valve (1)	Mono (1)	PIP-TZM, amoxiclav	[25]

* Patients included from our case series. Abbreviations: coagulase-negative staphylococci (CoNS), incision and drainage (I/D), penicillin (PCN), ceftriaxone (CTX), amoxicillin–clavulanate (amoxiclav), piperacillin–tazobactam (PIP-TZN), trimethoprim–sulfamethoxazole (TMP-SMX).

## Data Availability

Not applicable.

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
