# Peer review of "A 2-Year Retrospective Case Series on Isolates of the Emerging Pathogen Actinotignum schaalii from a Canadian Tertiary Care Hospital"

_microorganisms, 2022, doi:10.3390/microorganisms10081608_

Round 1

Reviewer 1 Report

The authors describe in a comprehensive manner Actinotignum schaalii case series with different points of infection. Overall it is interesting that these bacteria react to common antibiotics, which are rarely prescribed in patients above  60 years of age vs. metronidazole and ciprofloxacin. These antibiotics are often preferred in elderly patients. 

Minor comments: in Table 1, section 3, Urosepsis, there is a misprint "Uteric stone". Is it urethric or ureteric stone? 

The tables and figures are correct, and the conclusions are correct. 

Author Response

We appreciate the time and effort of the Reviewer in helping us improve our manuscript. As requested in Table 1, section 3, Urosepsis, the misprint is now changed to "Uteric stone".

Thank you

Reviewer 2 Report

This manuscript by Kakodkar et al reported a retrospective review on isolates of the emerging pathogen Actinotignum schaalii in a Canadian tertiary care hospital. Generally, the results could be valuable for physicians and other researchers in this field. However, there are still some critical issues to be addressed before further consideration.

Major issues:

1. The author claimed that “We also performed a literature review on Medline database from January 1997 to 67 November 2021”. But to my surprise, the newest reference in this manuscript was published in 2018. In reality, there have been many papers published in this area (check Google Scholar) over the past decade. Therefore, I recommend a thorough literature review should be reconducted.

2. This work involved a comparatively small number of patients, and I even can’t find any statistical analysis or statistical significance. Therefore, these findings reported here should be considered with extreme caution.

3. In this work, the clinical data of patients that grew A. schaalii from January 2020 to 2022 was documented. It is unclear if the COVID-19 pandemic can affect the reported results of this retrospective study? Specifically, it should be noted that hospital visits and admissions decreased significantly during the COVID-19 pandemic period. Why not prefer a time frame before COVID-19?

Author Response

We sincerely appreciate the time, effort, and insightful comments brought up by the reviewer to improve our manuscript.  The following are the responses for the major issues:

1. Medline shows only 5 more articles from 2018-2022 of which 4 have no usable primary clinical data for individual patients. There is only pooled patient data and/or missing clinical data. Data from this 1 article is now added to the urological literature review Table S2. Utilizing a database such as a google scholar, which has much more articles but most of them do not satisfy our inclusion criteria (ie. in English language and contains primary clinical data and not review articles). Additionally, adding the findings of google scholar is a laborious task and will not change the distribution of our findings significantly as we have data from 166 patients. Additionally, our ancillary goal was to provide an overview of the literature on A.schaalii and not to conduct a systematic review or meta-analysis.

2. This is a fair statement. There is no control group in this cohort hence no statistical tests were performed to report the significance of findings in each group. Only descriptive statistics were utilized.

3.  This is also a fair statement. It will now be added to the discussion section (: Line 228-230) that COVID-19 Pandemic could have affected the admission rates leading to a possible decrease in the incidence rate of A.schaalii detection. Our MALDI-TOF database did not include A.schaalii in earlier time periods, hence we could not search for an earlier time period.

Thanking you

Reviewer 3 Report

microorganisms-1824282

The manuscript of Kakodkar & Hamula is an extraordinary way of presentation of mixed review and own analytical case study type research. It is comprehensive manuscript and the aim looks really significant especially due to the polyinfections and antimicrobial tailoring of treatment.

The manuscript has been prepared carefully and is clearly organized. The only aspect that is confusing is the 55 publications from 1997-2021. It is worth mentioning how many of them were finally left after removing the publication on pooled data and those with no individual clinical data. I would also rewrite the introduction a little bit to show what is the scale of the problem.  I recomend to  accept the manuscript after minor revision.

Author Response

Thank you for your time and amazing insight. As requested this has now been added to Line 80-84. "55 publications from Medline, 30 publications were included in the literature review (166 patients). 12 articles were eliminated as there was no usable primary clinical data for each patient (it was pooled, or only had microbiological data). 7 articles were eliminated as they were review articles. The remaining 6 were not accessible." 

The introduction is now corrected Line 41-50 " Historically, the clinical underestimation of A. schaalii was linked to a lack of differentiation from Actinomyces and Actinobaculum spp., absence from MALDI-TOF MS databases, and suboptimal recovery and slow growth in routine cultures. Current clinical microbiology laboratories utilize matrix-assisted laser desorption/ionization time-of-flight mass spectrometry (MALDI-TOF) and its accurate bacterial identification repository. Despite this, A. schaalii infections remain underdiagnosed particularly when isolated in mixed culture scenarios. The role of A.schaalii in abscess formation is poorly understood. The absence of anaerobic culture media plates in routine urine workups also likely contributes to decreased recognition of this pathogen as a cause of UTI."

Thank you.

Reviewer 4 Report

Kakodkar P et Hamula C investigated a very interesting topic. On one hand, their study is another fact-approving statement that urine is not sterile and if investigated with prolonged schemes, some bacteria that are uncultivable using usual methods can be detected. On the other hand, their paper gave an additional view on some overlooked bacteria such as Actinotignum schaalii – the authors presented their personal data and also made a review of the similar data in the literature. The methods were appropriate and the results were presented and discussed very well.

Some minor comments:

Line 57-58 Please provide more information on the cultivation of A. schaalii or at least refer a study. The procedure is different from the usual microbiological urine investigation and it will be interesting to have a full explanation. Has the hospital laboratory applied this procedure for all urine cultures?

How can the authors be sure that this microorganism caused the infection, especially for abscesses and ulcers which have polymicrobial etiology? Finegoldia magna, Bacteroides are true pathogens and could be the sole cause of infection. Will the authors consider checking in future work for specific immune response – antibodies against A. schaalii? In my opinion this also can be discussed.

Table2 – what is meant by treatment strategy – is this treatment after microbiological investigation? Please provide data for the antibiotics that have been used. If it is the antibiotics that are shown in the next column, please explain this. How was the therapy altered after the microbiological investigation?

Table 3  and 4 It will be good to add the results of the antimicrobial therapy -  if some paper in the literature review provides it or if the authors have personal data perhaps?

Line 247,248 – This statement is speculation as the cases are too small in number. The sentence should be rephrased.

Author Response

Thank you for your time and amazing insight. 

Line 63 to 70 now states that " A.schaalii recovered in this study grew on either routine aerobic urine culture plates (blood agar and chromogenic agar), from aerobic or anaerobic blood culture bottles or on wound culture workups with brucella blood agar and blood agar incubated under both aerobic and anaerobic conditions. Urine cultures in our lab are not incubated anaerobically. A.schaalii  are not strict anaerobes, they are facultative anaerobes that prefer anaerobic growth conditions but can grow slowly under aerobic conditions. The lack of anaerobic culture workup in urines likely contributes to under identification of this pathogen as a cause of UTI"

Line 253 to 256 now states that, "Future studies could incorporate and correlate these microbiological findings with the antibody serology against A. schaalii to explore the pathogenic role of this organism in abscesses. The role of A.schaali in mixed infections is not always clear. As seen in our study, it can be co-isolated along with known pathogens in many cases".

Table 2 shows the treatment strategy before microbiological investigations. The treatment strategy includes surgical options and empiric antimicrobials prior to microbial culture results. The alteration of the antimicrobials after microbiological investigations is shown in table 3

The results of the antimicrobial therapy would definitely be interesting but nearly all included papers did not provide this data.

line 265-269 has now been rephrased “It is a possibility that both these procedures could help A. schaalii to form cystitis and pyelonephritis at sites beyond its native residence or in severe cases disseminate via the bloodstream in cases of septic joint infections, urosepsis, and endocarditis. Although given the small sample size this finding needs to be considered with extreme caution.”